



# Natural gas supply from Russia derived from daily pipeline flow data and potential solutions for filling a shortage of Russian supply in the European Union (EU)

Chuanlong Zhou[1], Biqing Zhu[1], Steven J. Davis[2], Zhu Liu[3], Antoine Halff[4], Simon Ben Arous[5], Hugo de Almeida Rodrigues[5], Philippe Ciais[1]

[1]Le Laboratoire des Sciences du Climat et de l'Environnement, Saint-Aubin, 91190, France
[2]Department of Earth System Science, University of California Irvine, Irvine, CA 92697, United States
[3]Department of Earth System Science, Tsinghua University, Beijing, 100190, China
[4]SIPA Center on Global Energy Policy, Columbia University, New York, NY 10027, United States
[5]Kayrros Inc., Paris, 75009, France

*Correspondence to*: Philippe Ciais (philippe.ciais@lsce.ipsl.fr)

**Abstract.** Russia is the largest natural gas supplier to the EU. The invasion of Ukraine was followed by a cut-off of gas supplies from Russia to many EU countries, and the EU is planning to ban or dramatically reduce its dependence from Russia. We provide a dataset of daily gas consumption in five sectors (household and public buildings heating, power, industry, and other sectors) with supply source shares in the EU27 & UK from 2016 to 2022. The dataset separates the contributions of Russian gas imports, and other supply sources, and accounts for storage to estimate consumption. The dataset was developed with a gas network flow simulation model based on mass flow balance by combining data from multiple datasets including daily ENTSO-G pipelines gas transport and storage, ENTSO-E daily power production from gas, and Eurostat monthly gas consumption statistics per sector. The annual consumption data was validated against BP Statistical Review of World Energy and Eurostat datasets. We secondly analysed the share of gas supplied by Russia in each country to quantify the 'gap' that would result from a cessation of all Russian exports to Europe. Thirdly, we collected multiple data sources to assess how national gaps could be alleviated by 1) reducing the demand for heating in a plausible way using the lower envelope of gas empirical consumption – temperature functions, 2) increasing power generation from sources other than gas, 3) transferring gas savings from countries with surplus to those with deficits, and 4) increasing imports from other countries like Norway, the US, and Australia from either pipelines or LNG imports, accounting for existing capacities. Our results indicate that it should be theoretically possible for the EU to make up collectively for a sudden shortfall of Russian gas if combining the four solutions together, provided a perfect collaboration between EU countries and with the UK to redistribute gas from countries with surplus to those with deficits. Further analyses are required to investigate the implications for the costs including social, economic, and institutional dimensions, political barriers, and negative impacts on climate policies with inevitable increases of $CO_2$ emissions if the use of coal is ramped up in the power sector.



## 1 Introduction

Russia is the largest natural gas supplier to the EU, where gas is used for households and public buildings heating, cooking and hot water production, power production, and industry (International Energy Agency, 2022). In 2020, EU countries consumed 155 billion cubic meters of natural gas from Russia, which represented more than one-third of their total gas consumption (Eurostat, 2022a). The invasion of Ukraine was followed by a cutoff of gas supplies from Russia to Bulgaria, Poland, and France. The EU is further planning to dramatically reduce its imports of gas from Russia (Mcphie et al., 2022). Articles published in the media show diverging estimates of the Russian gas dependence across the EU. These analysis lack high time resolution and detailed sector-based analysis (Mcwilliams et al., 2022a, b).

In addition to assessing the amount of Russian gas used in EU countries and its variation over time, it is also important to investigate how a shortage of this gas supply source could be alleviated. Significant reshaping of supply-demand structures of gas would be inevitable in case of a shortage of Russian gas, which could impact: 1) energy prices, economic growth and household income, 2) energy structure and environmental and climate goals, e.g. if countries seek to use more coal power (Eddy, 2022; Afp, 2022) to compensate for a shortage of gas or excessive prices, and 3) global energy markets and security, if the increasing demand of gas in the EU raises the global gas price.

To quantify the magnitude of the use of Russian gas in different countries and sectors, we present a new methodology based on daily data of pipeline gas flow, production, storage, and consumption of gas across EU27 countries and the UK. The data include daily pipelines gas flows across gas balancing zones of the pipeline network and storage facilities (Entsog, 2022a), daily power production from gas (Entsoe, 2022), and the monthly to annual partition of gas used to different sectors including households, commercial and public buildings, industry and power (Eurostat, 2022a, b, c). The supply-storage-consumption amounts and shares from Russian supply and all other supply sources, were calculated from the above data based on mass balance. We then investigate how a shortage of Russian gas equivalent to a complete stop of supply could be filled by reducing demand for heating, increasing power generation from other sources, increasing production in the EU, and increasing international imports both at LNG terminals and pipelines from non-EU countries other than Russia. We further consider existing transmission constraints on the intra-EU gas reallocation with the current pipeline infrastructure.

We provide two datasets, the EU27&UK daily gas supply-consumption (EUGasSC) with the share of different supply sources including Russia, and the EU27&UK daily gas reduction potential (EUGasRP). The EUGasSC data give the country- and sector-specific natural gas supply-storage-consumption at a daily resolution. These data allow us to quantify the shortfalls if Russian imports were to terminate. The EUGasSC data can be used for country- and sector-based policy decision-making and further socioeconomic analysis. The EUGasRP shows the daily gas consumption saving potentials that would be achieved by reducing demand for heating, increasing power generation from coal, nuclear, and biomass. Based on EUGasRP, we discuss whether demand reductions in heating, shifts in power generation towards nuclear and coal, and intra-EU and international coordination, particularly with the UK, the US, Australia, and Norway, could allow the EU to make up for a sudden termination of Russian gas imports.



## 2 Methods

### 2.1 Data collection

The workflow of this study is shown in the left panel of Fig 1. We collect several open datasets as input data : 1) ENTSO-G daily physical pipeline flow (Entsog, 2022a), which was used to simulate gas transmissions, consumption, storage and imports, 2) hourly ENTSO-E electricity generation (Entsoe, 2022; Liu et al., 2020), which was used to estimate how the Russian gas gap could be alleviated by increasing coal and nuclear  power (section **2.3.2**), 3) gas import and energy balance datasets from Eurostat, used to adjust/complete sectoral consumption values for ENTSO-G data and as cross-validation of annual consumption totals, 4)  BP Statistical Review of World Energy (Bp, 2022) to estimate the potential global increment capacity for LNG import and within EU production and as data cross-validation, 5) ERA5 daily 2-meters air temperature data (Copernicus Climate Change Service, 2019), which was used to estimate the potential reduction of gas consumption from heating sector based on consumption-temperature curves (section **2.3.1**). All the datasets were collected from APIs or manually download from websites.

### 2.2 Daily gas supply and consumption

To quantify country- and sector-specific gas supply and consumption, we built a graph network simulation model of daily physical gas flows for the period 2016-2022. The model simulates gas supply, temporary storage, and consumption sources for households and public buildings, power generation, industry, and other sectors in each country, as shown in the right panel of Fig 2. The detailed equations and the model are presented in SI. We completed the raw ENTSO-G data with Trading Hub Europe (THE) for German consumption (Trading Hub Europe, 2022), and e-control for Austria consumption (E-Control, 2022) as model input data. The simulation, in brief, evaluates the daily share of supply/consumption source of nodes (country or region) and edges (pipeline) by iteratively solving the mass balance of the physical gas flow in the network. We assumed that mass balance of each supply source is achieved daily for the transmission network and storage, so that the simulation results from the previous day are used as initial values for the next day. The gas consumption in the simulation was split into five sectors: household buildings, public buildings, power, industry, and other sectors based on the Eurostat energy balance datasets (Eurostat, 2022b, c). The simulated sector splitting values are validated with data reported by few counties where ENSTO-G data directly provide details on usage splitting for the distribution (DIS, covering heating and other sectors) and final consumers (FNC, covering power and industrial sectors) groups of sectors. The details are presented in SI. We performed the simulation from Jan 1$^{st}$ 2016 to Feb 28$^{th}$ 2022 for each EU27 country and the United Kingdom (UK) with a daily resolution, and separated the share of different supply sources (Russia, Norway, Algeria, Azerbaijan, Libya, Serbia, Turkey, LNG imports, and EU production) in the above listed consumption sectors, which yields the EUGasSC dataset.





### 2.3 Potential solutions to overcome a shortfall of Russian gas supply

The magnitude and temporal variation of a Russian gas supply shortfall, the gap, was diagnosed from EUGasSC as the share of Russian gas consumed each day. We then investigated the capacities of potential solutions that could fill in-country gaps or create surpluses in our country that could be re-allocated to fill gaps in other countries. Our goal is to estimate upper bounds for different solutions to alleviate the Russian gas gap, not to predict future mid-term changes in gas demand. The potential solutions considered include 1) reducing demand for heating, 2) increasing power generation from coal, nuclear, and biomass, and 3) rising international imports and European productions, as detailed below. The daily maximum potential capacities of gas saved of the first two solutions define the second dataset, EUGasRP. Note that we only investigated short-term solutions that could be immediately implemented e.g. for the upcoming year, given strong assumptions: that the gas supply for the industry will be prioritized and remains at current levels, that no massive increase in power production from renewable energy will happen in the next year, although long term investments could partly substitute Russian gas use by renewables.

### 2.3.1 Reduced gas use in residential households and public buildings sectors

The potential gas consumption reductions for reduced heating in buildings were estimated for each country based on empirical temperature-gas-consumption (TGC) curves, similar to those shown by (Ciais et al., 2022). The TGC curves were constructed based on daily consumption from EUGasSC and daily population-weighted air temperatures based on the Eurostat population dataset (Eurostat, 2022d) and ERA5 daily 2-meters air temperature data (Copernicus Climate Change Service, 2019). Fig 2 shows an example for France, with the TGC curve fitting (Fig. 2a), how the consumption reduction was estimated (Fig. 2b), and time series of reduced gas consumption (Fig. 2c). The TGC curves were fitted with a two-segment linear regression separated by a critical temperature (the start-heating temperature), as shown in Fig 2a. Then we constructed plausible reduction scenarios, as shown in Fig 2b, by modifying two parameters of the TGC curves: 1) assume a lower start-heating temperature, and 2) compute a plausible lower slope below the critical temperature, the slope representing the increase of gas consumption per unit of air temperature decrease. Lower slopes were estimated using only data below a low threshold percentile of the observed consumptions data. Flatter the slopes and the larger the gas consumption savings for heating can be achieved with lower thresholds. Finally, the actual reductions in daily consumption were calculated as the difference between the original and the modified TGC curves, as shown in Fig 2c. Similar figures plots for building consumption reduction in other countries and for other reduction parameters are presented in SI.

We build a moderate and a drastic reduction scenarios for gas saving in household and public buildings as follows: 1) households on weekdays adopt a 1 °C lower critical start-heating temperature (2 °C for the drastic case) and using the lower 30[th] percentile of TGC curves to define the slope (the 20[th] percentile for the drastic case), 2) households on weekends adopt a 1 °C lower critical temperature (2 °C for the drastic case) and the lower 50[th] percentile of TGC curves (40[th] percentile for severe case) based on the assumption that heating gas consumption is systematically lower during weekends compared to



weekdays, and 3) public buildings adopt a 2 °C lower critical temperature (4 °C for the drastic case) and the lower 30[th]
percentile of the TGS curve (20[th] percentile for the drastic case).

### 2.3.2 Reduced gas use in the power sector

The gas consumption reduction in the power sector was estimated by substituting gas by coal, nuclear, and biomass. This
assumes that EU coal producing countries like Germany and Poland will be able to increase their coal production or that coal
imports will be increased. Oil was not considered as an alternative fuel because Russia is also the largest oil supplier to the
EU, although some gas-fired power plants can easily switch to oil. To evaluate the capacities of gas consumption reduction in
the power sector, we assumed that the electricity generated with gas can be substituted by boosting hourly electricity generated
with coal, nuclear, and biomass up to a maximum level defined by recently observed data since 2019. We estimate this
maximum level as 75% of the maximum observed hourly power generation capacities for coal, nuclear, and biomass of each
country for a moderate gas reduction scenario (95% for a drastic reduction scenario), based on observed ENTSO-E electricity
production data from 2019 to 2021 (presented in SI). The capacities of each alternative power supply sources are estimated
from the hourly difference between actual electricity generation and the maximum assumed level. Finally, we aggregate hourly
coal, nuclear and biomass power capacities to daily resolution and convert them to an equivalent reduction of gas consumption
using an average gas power plant efficiency for each country. Those efficiencies are estimated based on regressions between
gas consumed by final consumers (from ENTSO-G) and gas-powered electricity (from ENTSO-E), as presented in SI.

### 2.3.3 Increased supply from import and EU production

Potential increases of  LNG imports, pipeline imports, and production within the EU27&UK were estimated based on the BP
world energy report (Bp, 2022). To do so, we calculate maximum supply (imports or production) values by comparing: 1)
historical maximum capacity in a list of countries that could export gas to Europe, in the period 2010 - 2020, and 2) recent
increment capacity, which equals to 2020 value × 2020 growth rate. We only included capacities of increased supply by
selected supplier countries. For LNG, these are the United States, and Australia. For pipeline imports, supplier countries are
Algeria, Norway, Azerbaijan, and Libya. For increased domestic production, we considered the Netherlands, United Kingdom,
Romania, Denmark, Germany, Italy, and Poland.

### 2.3.4 Intra-EU transmission constrains

Some EU countries can reduce their gas consumption not only to alleviate a domestic shortage of Russian gas but can also
generate a surplus of gas, which we assumed could be transferred to other countries with a deficit, i.e., those that could not
fully alleviate a shortage of gas from Russia. This implies to consider transmission constraints on the intra-EU gas reallocation
based on pipeline directional capacities given by ENTSO-G (Entsog, 2022b). We performed gas redistribution simulations as
described below to evaluate the fraction of the Russian gas gap that could be alleviated at EU scale by intra-EU gas transmission
from surplus countries to deficit counties. The gas redistribution simulation was performed by modifying the model described



in section **2.2** as follows: 1) adding the estimated capacity/gap to each node, 2) constraining the pipeline transmission capacities
for the edges, 3) creating redistribution flows if nodes have extra capacity and the connected pipelines have extra capacities to
transmit gas, 4) solving the maximal redistribution capacities in the network based on ENTSOG flow and redistribution flows.
Then the transmitted surpluses or deficits for each country are be calculated after the redistribution simulation.
The transmission by the current ENTSOG gas pipeline network is highly directional from Russia towards EU countries. For
example, there is a large transmission capacity (614 GWh/day) from Germany to France, however, with zero capacity from
France to Germany due to different systems for odorized gas (Entsog, 2022c, b). In this case, we assumed that pipeline
directional flow could be still fully reversed in the network, although, in reality, such a fully reversed flow scenario for the
current infrastructure remains uncertain a short-term period (Entsog, 2022c).
**3 Data validation and uncertainty estimation**
We validate the EUGasSC dataset with Eurostat datasets (Eurostat, 2022a, b, c) and BP Statistical Review of World Energy
for the following variables: 1) annual total gas consumption (Fig 3a), 2) monthly total gas consumption (Fig 3b), 3) annual
total LNG imports (Fig 3c), 4) annual total EU gas production (Fig 3d), and 5) total gas consumption in each country (Fig 3e).
The results show low discrepancies for the annual total consumption (12±5% with ± being the standard deviation across years
or all EU countries), monthly total gas consumption (11±7%), annual total LNG imports (0±14%), and total gas consumption
in each country (9±7%, excluding Spain and Latvia-Estonia), whereas large differences were found for the annual total EU
gas production (-42±12%), Spain (-65%) and Latvia-Estonia (-153%). A negative difference means that our dataset has lower
values than Eurostat or BP data. The validations of our dataset with Eurostat are done for the total consumption, even though
Eurostat was used for splitting the consumption sectors in EUGasSC (section **2.2**). Thus, the use of Eurostat data for
consumption attribution and the national total cross-validation is not circular in our approach.
The larger differences between EUGasSC and Eurostat were found for the year 2020, because the UK data were not provided
in Eurostat dataset due to the Brexit. Although our validation results indicate an overall good quality of our dataset,
uncertainties still exist: 1) we might underestimate the EU production. As Fig 3d shows, EUGasSC has significant smaller
production values compared with both Eurostat and BP datasets, 2) the consumption differences in each country might bias
our analysis of potential solutions for the Russian gap particularly in Latvia-Estonia, as our dataset underestimates their gas
consumption. Other uncertainties from collecting, processing, and analysis mainly arise from the facts that: 1) values and
figures in this paper follow the ENTSO-G data collected in April 2022, but ENSTO-G regularly correct and update their
database even for very early data (we will also update the EUGasSC and EUGasRP datasets regularly), 2) we estimate the
daily Russian supply share based on a simulation that assumes a daily balance of the pipeline network, which might over-
simplify gas balancing processes, 3) our estimation of daily sectoral consumption have uncertainty as split the consumption
sector based on monthly consumption variation pattern and Eurostat energy balance for those countries that do not report daily
gas consumption attribution to ENSTO-G, 4) we estimate potential solutions for alleviating Russian supplies gap based on



empirical capacities with a number of assumptions, without considering social, economic, international cooperation, and geo-
political barriers.

## 4 Results and discussions

### 4.1 Sectoral and country-based differences in Russian gas consumption

The sectoral and country-based gas supply-consumption patterns for the EU27&UK are shown in Fig 4. For 2021, the sectoral
gas consumption in the EU27&UK in decreasing order are household heating (1677 TWh, 29% of total) > industrial (1648
TWh, 29% of total) > power (1648 TWh, 22% of total) > public building heating (672 TWh, 12% of total) > others (461 TWh,
8% of total). Consumption patterns of Russian gas are highly country-dependent. The five biggest Russian gas consumers in
2021 are Germany, Italy, Hungary, Poland, and Austria, which together consumed 77% of total Russian imports. Considering
their relatively high Russian gas share (from 53% to 89%), obstacles to alleviating the Russian gas gap might be serious in
those countries. On the other hand, France, Netherlands, Belgium-Luxembourg consumed altogether relatively large absolute
amounts of Russian gas (40% of the Russian supply excluding the largest five countries), but gas from Russia nevertheless
represents a small relative share of their total gas consumption (from 12% to 19%). Southern and Northern European countries
that are close to Russia, including Czechia, Slovakia, Croatia, Slovenia, Denmark, Sweden, Finland, Latvia, Estonia, and
Lithuania, consume less Russian gas in absolute amounts due to small country and population sizes, however, but have large
shares (from 56% to 92%). The rest of the countries, including Romania, the UK, Spain, Ireland, Bulgaria, Portugal, and
Greece, use a small absolute and relative amount of Russian gas. Those results suggest that solutions and the difficulties of
shifting energy supply sources and resolving the gas supply gap can be significantly different among the EU27 &UK.
Therefore, we combined countries with similar patterns and closer distances together when discussing potential solutions in
the following sections.

### 4.2 Gas supply shares and recent trends

We analyzed the gas supply shares and trends for imports from Russia (RU), Norway (NO), LNG imports (LNG), other imports
through pipelines connected to EU such as from Azerbaijan and Algeria (Other), and EU production (PRO) from Jan 1st 2019
to February 28st 2022 at the breakout of Russian invasion of Ukraine, based on our EUGasSC dataset. As shown in Fig. 5, we
found that there was a relatively constant gas supply structure before 2021. Annual changes for all the supply sources were
ranging from -1.6% to 1.6%. The supply shares before 2021 in decreasing order are Russian (36±4%) > Norwegian (26±3%)
> LNG (21±4%) > EU production (10±2%) > other exporters (5±2%). However, significant changes of supply sources occurred
after 2021, particularly for Russia (decreased by 11%), LNG (increased by 9%), and Norway (increased by 4%). The EU gas
price (Fig 5, top panel) shows three distinct values: 1) relatively constant before 2021 suggesting a stable gas supply structure,
2) gradually increasing from the start of 2021 to the middle of 2021 with a shift of gas supply sources, and 3) a sudden peaked
and high variability on top of at a high plateau as tensions over the situation in Ukraine increased.



The daily gas consumption and Russian supply share for 2021 is presented is Fig 6. The gas consumptions are systematically
lower on weekends and during warm seasons (from May to October) due to less heating and industrial requirements (Fig 6,
top panel). Although it varied with the total gas consumption, the Russian supply declined less than the total gas consumption
in the warm seasons when the EU imports and stores gas at a lower price (Fig 6, middle panel). Rapid Russian gas supply
changes can be observed between the first half of 2021 (Fig 6, left end of bottom panel) and the second half of 2021 (Fig 6,
right end of bottom panel). Comparing the Russian share in 2021 with previous years, the EU27&UK had a higher reliance on
Russian gas supply in the first half of 2021, but this reliance sharply declined at the end of the 2021.

### 4.3 Russian gas gaps

The magnitude of country-level and regional shortfalls in Russian gas supplies if imports from Russia were to terminate are
shown in Fig 4c, and summarized in Fig 7 (hatched red bars). In 2021, the total consumption of Russian gas in the EU+UK
was of 2090 TWh, corresponding to 36.6% of total gas consumption. Germany and Italy consumed the largest quantities of
Russian gas, accounting for 52.4% of all Russian gas consumed in the EU+UK (1096 TWh). Less Russian gas was consumed
in Hungary, Poland, and Austria, as well as in Baltic and Nordic countries and other central European countries (Slovakia,
Croatia, Slovenia, and Czech Republic, totalling 768 TWh). Russian gas nevertheless represented the dominant share of gas
used in these countries (77.0%). The UK and other EU countries (Ireland, Bulgaria, Portugal, and Greece) have smaller
dependencies on Russian gas in both absolute and relative amounts.

### 4.4 Potential solutions

We present the country-based capacities for alleviating the Russian gas gap with a daily resolution from the EUGasRP dataset.
These potential solutions (section **2.3**) include demand side reduction for household and public buildings, increasing power
production from coal, nuclear, and biomass, and increasing EU production as well as imports. An overview is presented in Fig
7. We found that, according to our demand reductions scenarios, saving gas for heating in buildings could cover 17%~23% of
the total Russia gap. TGC curves and reduction estimation for each country and scenario are presented in SI. An additional
fraction of 18%~41% of the total Russian gas gap could be achieved by substituting gas with coal, nuclear, and biomass sources
for power generation. Increasing coal-fired power would save 218~497 TWh of natural gas, and nuclear power another
142~317 TWh. France alone in our scenarios contributes 40%~51% to nuclear power capacity in this scenario. The uncertainty
ranges of the solutions are estimated from the different scenarios presented in section **2.3.1** for the heating sector and **2.3.2** for
the power sector. On the supply side, we estimate from recent data on production and production change (section **2.3.3**) that
increased natural gas production in EU countries and the UK could fill up only 5% of the gap. Note that, however, our dataset
might underestimate European gas production as discussed in section **3**. We further estimate that US and Australia might be
able to produce up to an extra 470 TWh LNG, and the rest of the world might be able to produce up to an extra 414 TWh LNG,
and other exporters including Norway might produce up to an extra 115 TWh of gas carried by pipelines, based on section
**2.3.3**. Those extra international supplies, on top of reduced heating consumption and increased power form non-gas sources





would be sufficient to cover almost entirely the remaining gap, leaving less than 4% of the total Russian gas gap. This might
entail substantial changes in the global gas market and LNG prices, and potentially exacerbate economic inequalities in the
EU and globally, which is outside the scope of this study.

### 4.5 Challenges and uncertainties

Our estimates do not contend with social, economic, and political barriers, from the international gas/LNG market. Our
proposed solutions are highly country-dependent gaps. For example, some countries can easily overcome small shortfalls in
Russian gas, the less Russian-gas dependent countries in Fig 7, while other countries might be able to use less gas because of
their particular energy structure, e.g, France may switch to more nuclear power. But Germany, Italy, Austria, and Hungary
cannot readily replace Russian gas. Our analysis assumed a perfect cooperation between EU27 members, the UK, as well as
with the United States, Australia, and Norway, to maximize the gas consumption reduction, production, exports, and optimally
redistribute the gas surplus. However, such perfect cooperation might be vulnerable to unforeseen events such as the recent
gas workers' strike in Norway (Harrington and Cooban, 2022). Cooperation within EU can be affected by other competing
factors, such as gas needs from other regions (e.g., Japan) being also affected by a shortage of Russian gas supply (Energynews,

263 2022).

The solutions presented in this study, also assume that countries that could generate more power without using gas could fully
transfer their gas surplus to those having gaps, without other constraints of intra-European pipeline than current transmission
capacities. However, optimally redistributing gas from countries with surplus to those with deficits could be another barrier.
Only 85 TWh capacity could be transmitted to Russian-dependent countries with the current pipeline infrastructure network
(light red in top panel in Fig 7), which would leave 1094~1624 TWh of Russian gap (19%~28% of the total gas consumption).
This gap is much larger than the extra gas we estimate could be brought in from the global market. The major issue causing
transmission limitations is the current pipeline directions. For example, the total remaining gap could be reduced up to 844
TWh (dark red in top panel in Fig 7), which could be resolvable by global LNG imports only if transmission could be redirected
from France to Germany, while the current transmission capacity is from Germany to France (details see SI).
Increasing nuclear power back to the high levels of the last years may also be challenging. Germany may not reopen or boost
output from nuclear power plants (World Nuclear Association, 2022), and the current nuclear capacity in France is much
smaller than its designed capacity due to the routine maintenance or defaults detections of the reactors (Association, 2022;
Seabrook, 2022) . There are currently 12 nuclear power reactors in France out of a fleet of 56, being offline and inspected for
stress corrosion (World Nuclear News, 2022; Edf, 2022a), and there are 15 more reactors as reported by media not supplying
energy this summer because of regular maintenance (Parisien, 2022; Edf, 2022a). Those shutdowns of the nuclear reactors in
France resulted in a significantly smaller cumulative output in June 2022, 15.2% smaller than that in 2021(Edf, 2022b), which
might become an important limitation for filling the Russian gas gap in the EU as we estimated that France can create
considerable gas surplus by switching from gas to nuclear power. Last but no least, options to increase coal use, although
supported by some recent political declarations, may jeopardize the emission reduction targets of the EU if it was sustained



for several years (Afp, 2022; Eddy, 2022). We estimate that our scenario of increased coal power would result in an additional
70~159 MtCO2 emissions per year, which are equivalent to 3%~6% of total EU fossil $CO_2$ emissions in 2020 (Statista, 2022).

## 5 Conclusions

We presented two datasets for EU27&UK at daily resolutions: 1) the EUGasSC dataset describing the sectoral and country-
based daily natural gas supply-consumption, and 2) the EUGasRP dataset describing the daily sectoral and country-based
natural gas reduction capacities for the heating and power sector, increased EU production, and foreign imports. We used these
two datasets for analyzing the gas supply-consumption patterns and trends, quantifying how the Russian gas gap could be
alleviated if Russian imports were to terminate. Our results indicate that a full and sudden loss of Russian gas for the EU could
be theoretically filled with short-term solutions including plausible demand reductions in heating, higher power generation
towards nuclear and coal, and intra-EU and international coordination, particularly with the UK, the US, Australia, and
Norway, albeit with numerous challenges and uncertainties. For future research, the two datasets can be applicable to various
fields and topics, such as gas/energy consumption and market modelling, carbon emission and climate change research, and
policy decision making.

## 6 Usage note and data availability

We published the two datasets (EUGasSC and EUGasRP) as CSV files and hosted within the Zenodo platform at
https://doi.org/10.5281/zenodo.6833534 (Zhou et al., 2022). The datasets are open access, licensed under a Creative Commons
Attribution 4.0 International License. The column headings of the data dictionary files along with the unit of each variable are
listed in SI.
Our datasets provide daily gas supply-storage-consumption of five consumption sectors for the EU27&UK, and the potential
gas reduction capacities from heating and power sectors of each country as solutions for resolving Russian gaps. They can be
used as either input or reference datasets for further research of various fields, such as gas/energy modeling, carbon emission,
climate change, geopolitical policy discussions, and the international gas/energy market. The first author who collected the
data and performed the analysis and the corresponding author who is an expert on the background of this study is at the disposal
of the researchers wishing to reuse the datasets.





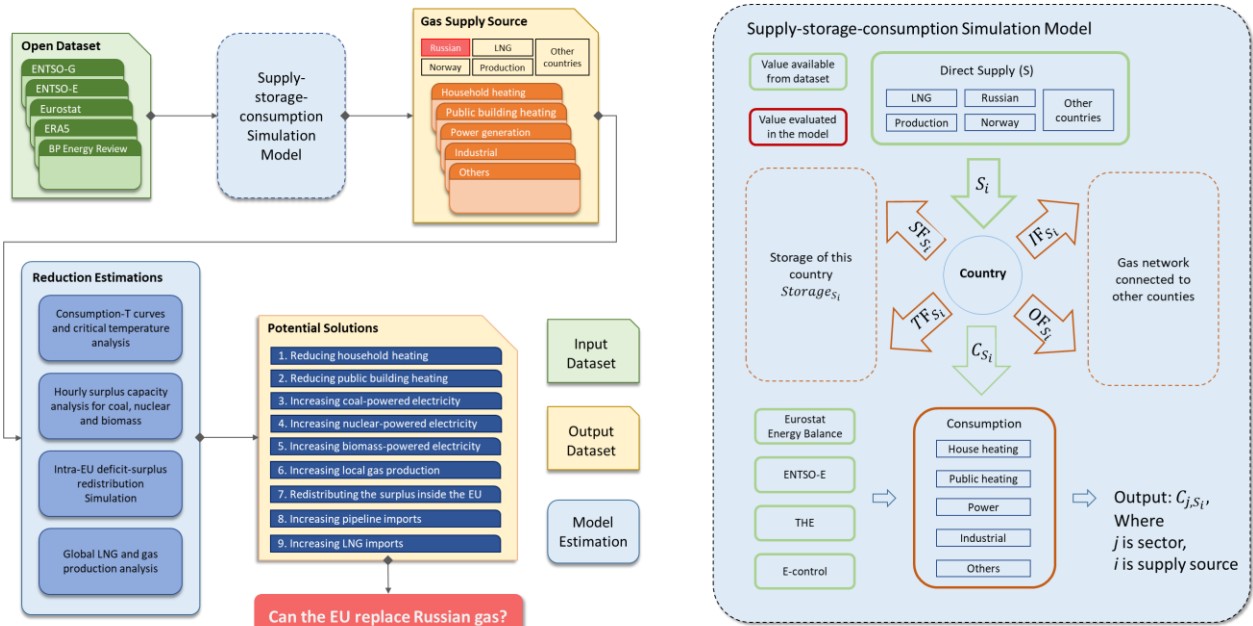

**Figure 1. Workflow and model concept of this study.** The workflow of this study including input dataset, their usage in models, and output datasets (left). The concept of supply-storage-consumption simulation model used in this study (right).

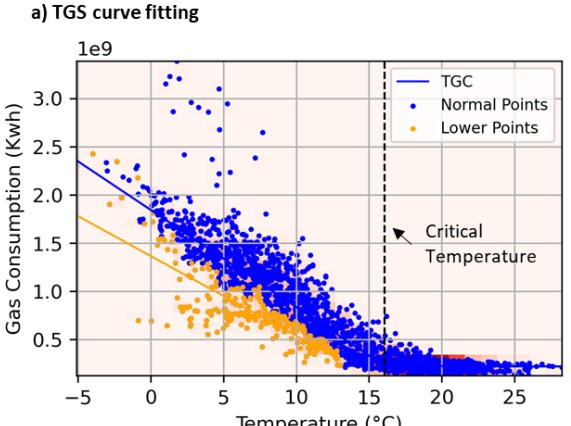

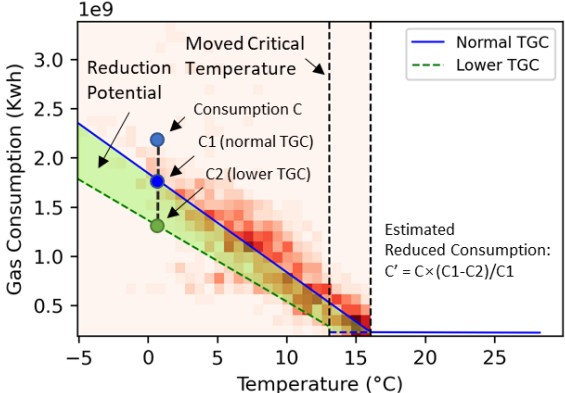

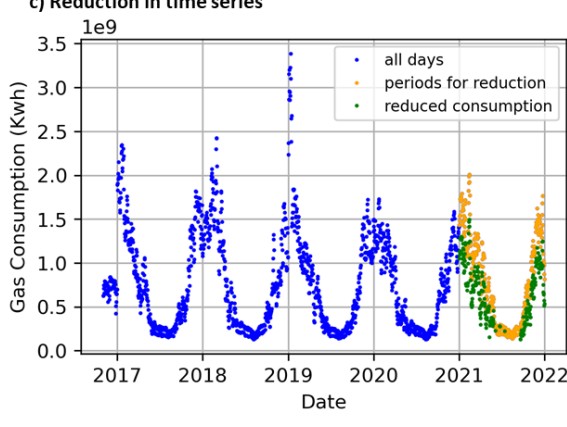


**Figure 2. Example of temperature-gas-consumption (TGC) curves and estimated reduced consumption.** The figures show the example
of house heating reduction estimations for France, a) TGC curve fitting for the normal consumption and lower 20% percentile consumptions,
b) how the reduced consumptions were estimated for each daily data point, c) the estimated heating reduction time series from 2021 to 2022.


**Figure 3. Data comparisons among dataset in this study, Eurostat, and BP Statistical Review of World Energy.** The figures show the
comparisons for a) total annual consumptions, b) total monthly consumption, c) total annual LNG import, d) total annual EU gas production,
and e) total country-based consumption from 2017 to 2022.





**Figure 4. Gas supply and sectoral consumptions in each country for 2021.** The figures show the country-based data for a) the sectoral consumption amount with Russian supply amount, b) the sectoral consumption share with Russian supply share, and c) the supply source amount (the inset plot). The countries are sorted by the amount of Russian supply.

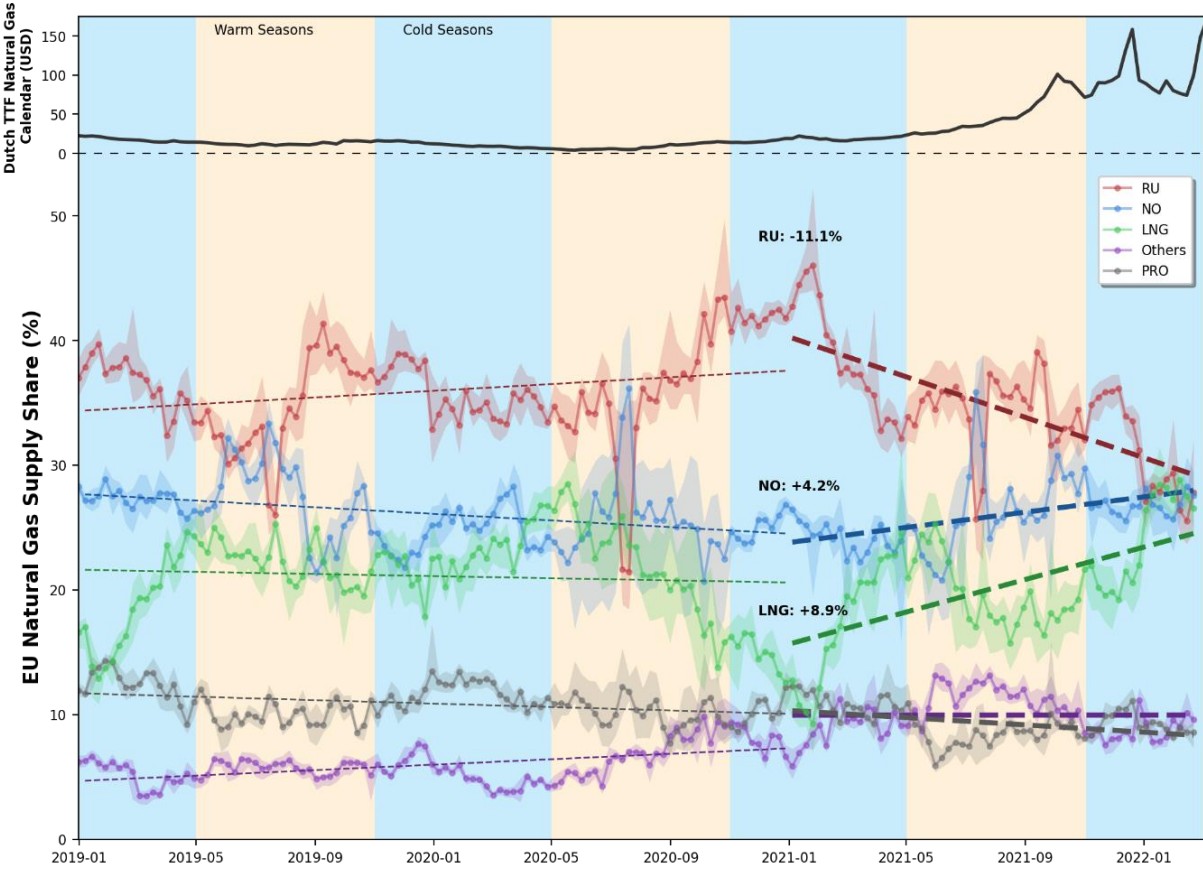

323

**Figure 5. Weekly natural gas supply share trends in EU27&UK with EU gas price.** The top figure shows the Dutch TTF Natural Gas
Calendar price as the EU gas price, and the bottom figure shows the weekly natural gas supply shares and trends for Russian imports (RU),
Norwegian imports (NO), LNG imports (LNG), other imports (Other), and EU production (PRO). The linear trends of different supply
sources for the periods from 2019 to 2021 and after 2021 (show as dashed lines). The confidential interval shows the variations of the week.



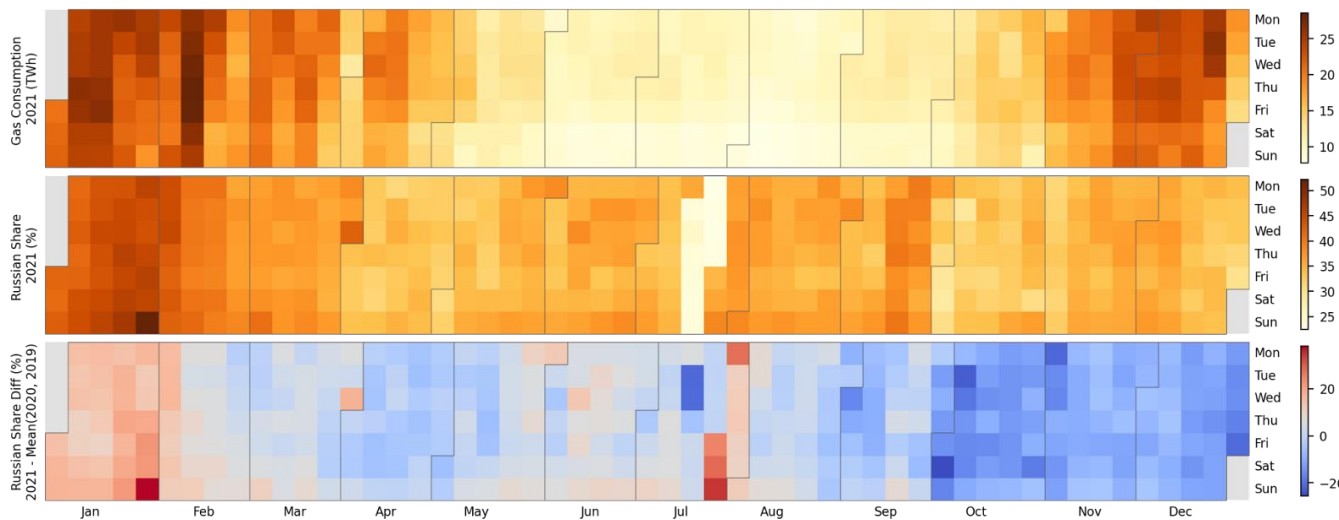

328

**Figure 6. Calendar plot of 2021 for the gas consumptions and Russian supply shares in EU27&UK.** The figures show the calendar plot
(each box represents a day and each column present a week) for gas consumption in EU27&UK (top), mean Russian supply share (middle),
and the difference of mean Russian supply share between 2021 and the previous two years (bottom).

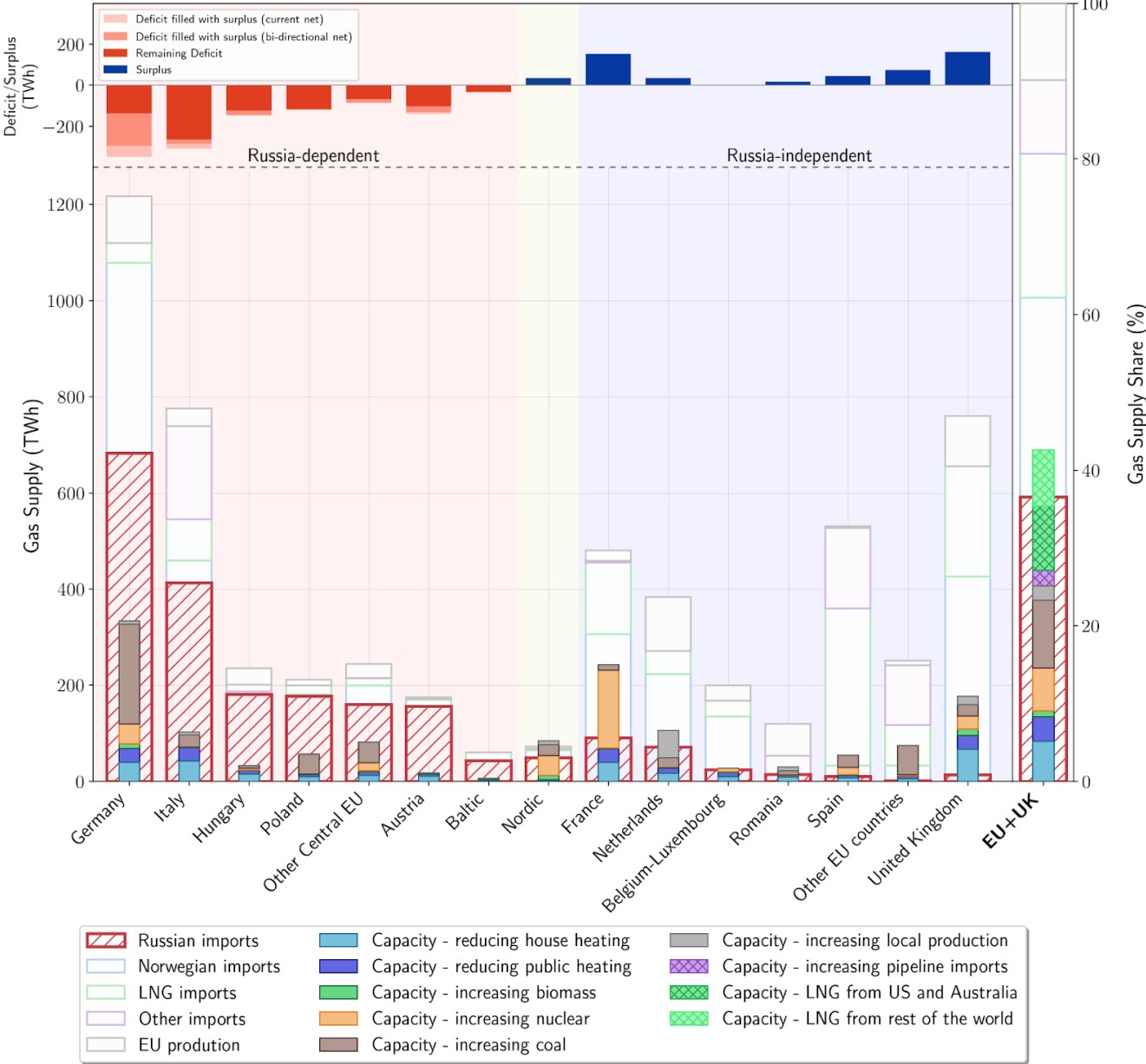

**Figure 7. Russian gas gap and potential solutions in the EU.** The wider bars are how the gas supply from Russia (in hatched red), Norway, LNG imports, other pipeline imports, and EU gas production. The narrower bars present the maximal capacity from different sectors to potentially fill the gap (see text). The EU+UK gap is presented as a percentage of total consumption (with the right y-axis). The top bars present the deficit and surplus, and the amount can be transferred inside the EU. 'Baltic' includes Estonia, Latvia, and Lithuania. 'Nordic' includes Denmark, Sweden, and Finland. 'Other central EU countries' includes Slovakia, Slovenia, Czech public, and Croatia. 'Other EU countries' includes Ireland, Bulgaria, Portugal, and Greece. The Russia-dependent countries have high Russian gas shares (>20%) with remaining gaps. Russia-independent countries have low Russian gas shares (<20%) with no remaining gaps. Nordic countries have higher Russian gas shares but no remaining gap.



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
