# Peer review of "Natural gas supply from Russia derived from daily pipeline flow data"

_Earth System Science Data, 2022_

## Referee Comment (RC1)

Review ESSD-2022-246

Many small text errors (typos, singular/plural, tense errors, etc.), too many to mark. One hopes very good proofreaders from Copernicus will catch most of those? If making changes in response to reviewer comments, authors could engage native English speaker to make small corrections?

Authors frequently refer to 'SI' by which they mean 'Supplemental Information'. For Copernicus journals, however (ESSD included) SI more commonly refers to Special Issue. These authors should clarify by making easy substitution of 'Supplement' for every use of 'SI'?

Line 139: This refers to Table S1? Authors could / should improve reader understanding by specifying which Supplement Figure or Table in each mention in main text.

Line 161: "France to Germany due to different systems for odorized gas (Entsog, 2022c, b)." Presumably, 'odorized' refers to safety requirements, e.g. different countries impose different chemical addition requirements that serve to inhibit opposite exchanges?

Lines 162, 163: Authors raise interesting potentially-troubling factor: flow irreversibility. Readers needs better understanding of magnitude and frequency of this barrier? Occurs at every 'upstream' (back toward Russia) node? Only in flows across specific edges? Close look at Figure S2 clearly implies some 'loops': how do or would mono-directional flows work in those cases? Perhaps a serious barrier? Topic gets worthy mention later (e.g. line 270 in Challenges and Uncertainties section (good) but again sans quantification?

Line 183, 184: "assumes a daily balance of the pipeline network, which might over simplify gas balancing processes," not clear how this possible oversimplification would impact overall calculations? Could authors provide an estimate? Would these errors prove systematic (e.g this work always lower or higher than validation products) or would they prove random? Valid point but not clear about impact?

Line 184, item 3): awkward, not clear what the authors intend?

Lines 186-188: valid concern, credit the authors for mentioning social geopolitical issues. Again, do authors expect these to prove negligible? Serious? Disqualifying? Pertinent to this topic but hardly unique to this particular approach?

Line 194: a bit confusing, I think authors mean 'largest consumers of Russian gas'?

Line 201, 202: countries with "large" shares but with small domestic consumption presumably pass through much of their RU gas? E.g. large imports but equally large exports? How does reader check this in EUGasSG? Not clear. One needs both EUGasSG and EUGasRP?

Line 205: Necessity to group like countries very clear, but this represents another further source of uncertainty? Not itemized above or below?

Line 215 and following: Interesting approach. Readers must assume, or need assurance (?) that Dutch TTF price data are open and easily accessible? These data merit listing in text and figure itemizations of important data sources?

Line 234, Section 4.4: Good discussion, follows directly (and, necessarily) from prior data collection and analysis, but - as projections - here we really need uncertainties? Authors could preface entire section with a disclaimer about, e.g plus/minus 10%, 20%. Something?

Line 253 This reader misses in this section a summary of strengths, uncertainties, and validations of EUGasSG data? Promote your work? What new have you discovered and provided, with what strengths and weaknesses. Otherwise, this section moves directly to gap analysis without assuring readers that we start from a good fresh skillful basis?

Line 254: Fair enough, but applies not only to international LNG sources?

Line 255: "country-dependent" yes, but earlier authors informed readers of necessity of combing groups of countries based on population size and proximity to Russia. Do those earlier groupings no longer apply here? Or, should authors add something about modifying country-by-country dependencies?

Line 260: If (admittedly) we rarely achieve "perfect cooperation", and if even best cooperation might prove vulnerable to distinct within-country events (e.g. strikes), don't these real-world issues introduce a further degree of uncertainty? At this summary of skillful analysis, authors owe readers a word of advice on how much to trust this work? Please do not let readers make their own assumptions?

EEGasSG easy to download, open and use but many data show 15 or more significant figures? Not believable, particularly given flow and sector uncertainties. This represents a float vs int problem? Unfortunately, authors reduce their credibility by expressing their data to obviously false precisions?

References: Confusing appearance? Some names in all-caps (e.g. 370, 372), others standard? Not clear the alphabetical organization in too many cases, e.g. Line 343 why Nuclear … here, Line 382 why 'EDF …' here, etc.

Line numbers absent in Supplement so hard to comment. But, under Supplement 'Power sector' section, reader finds (two lines below): analyze the diurnal capacities (75% and 95% as moderate and severe cases) for those alternative electricity sources. Properly, 'diurnal' refers to daily as 'nocturnal' refers to nightly; for a full 24-hour period the authors should specify 'diel'? Not a common usage but more correct than current?

---

## Referee Comment (RC2)

**Review of ESSD paper**

*Title: Natural gas supply from Russia derived from daily pipeline flow data and potential solutions for filling a shortage of Russian supply in the European Union (EU)*
*Author(s): Chuanlong Zhou, Biqing Zhu, Steven J. Davis, Zhu Liu, Antoine Halff, Simon Ben Arous, Hugo de Almeida Rodrigues, and Philippe Ciais*
*MS No.: essd-2022-246*

**General comment of appreciation**

This paper is not only documenting timely information but entails also important messages for policymakers to understand different options to address the current energy crisis with options at a given cost, also in terms of GHGs. This paper is recommended to receive extra advertisement of ESSD for two reasons. Firstly, it illustrates the importance of data and observation-based evidence for informed decision making. Substitutes for Russian gas supply are needed now and this paper comes with very timely advice. Secondly, science and data sharing can help to keep an open dialogue between many countries in troubled times. It is remarkable how much regional data has been taken up in this exercise.

**Comment on the data used for section 2.3.3:**

It is difficult to assess if the data and information collection is complete for Europe, given the very different approaches in the different EU27 MS. Import from North Africa might benefit extra attention. As an example, ENI Italy was already end April seeking to untap North-Africa's potential for gas supply, getting deals with Algeria and Egypt.

Ref: Italy looks at Mediterranean for alternatives to Russian gas – EURACTIV.com; Italy's ENI to tap North African gas for Europe - Al-Monitor: Independent, trusted coverage of the Middle East; pr-capital-markets-day-2022-eng.pdf (eni.com)

**Comment on the substitution discussed in section 2.3.2:**

Why is biogas production within EU27 not taken up? Even if the source is small, a little upscaling is still possible and all small bits help.

Ref: EU rolls out plan to slash Russian gas imports by two thirds before year end – EURACTIV.com; Biogas made from farm waste could replace Russian fossil fuels in Germany | Euronews

**Comment on Supplementary Information – gas network simulation**

The total amount of the gas pipelines from North Africa seems reduced. This might be acceptable, if the total volumetric flow rate is fully taken up. In fact, there are three pipelines available from North Africa to Italy: the Transmediterranian (via Tunesia – El Haouaria), the Greenstream (Libya – Mellitah) and the Galsi (Algeria – Koudiet Draoucha) and there are two pipelines available from Algeria to Spain: the Medgaz (Algeria – Beni Saf) and Maghreb-Europe gas (via Morocco – Gibraltar)

Ref: (15) Sustainable capacitated facility location/network design problem: a Non-dominated Sorting Genetic Algorithm based multiobjective approach | Request PDF (researchgate.net)

**Comment on Supplementary Information – sectoral splitting validation**

It would benefit the transparency, if EuroSTAT sectors 4 "Industry – E and NE" and sector 5 "sum of the other four sectors" could be described shortly. It would be interesting to understand if "fertiliser production" is part of sector 4 as Industry NE or of sector 5.

**Minor editorial comments:**

l.2 + l.36: please replace "dramatically" by "drastically"

l.35: More countries saw a sudden decrease and this sentence might be completed with a date and a reference.

l.25 + l.62: when mentioning USA, Australia and Norway, adding North Africa could be appropriate

l.179: please specify the reason why gas consumption for Latvia and Estonia are underestimated.

l.246: adding North Africa could be appropriate.

l.264: first verb "could" seems too much for good reading

fig.1: the "legend" figures for input dataset, output dataset and model estimation is not needed and could be deleted, if the first green box reads "Open input dataset"

fig.2: a) I would prefer to read "annual consumption" instead of "total consumption"

Fig.5: please write TTF in full

Fig.6: this is the difference of two shares, but for which the total maximum is different because of different years. I would prefer to see the difference between 2021 and 2020 (in absolute terms of Russian gas supplied) divided by the total in 2021 (of gas supplied). This would present the change in Russian supply share for 2021 compared to a previous year in a cleaner way. And since 2020 is a special COVID year, a fourth row would be welcome, where also the difference between 2021 and 2019(in absolute) divided by the total in 2021 is also presented.

Fig.7: excellent graph!

---

## Author Comment (AC1)

Dear reviewer,

Thank you very much for your valuable comments.

Here are our responses to the comments. The text with underlines is your original comments.

Overall

**We will perform proofreading and try to catch all the typos. Also, we will replace 'SI' with 'Supplement' in our paper for Copernicus journals.**

Line 139: This refers to Table S1? Authors could / should improve reader understanding by specifying which Supplement Figure or Table in each mention in main text.

**This refers to Table S1. We will modify the text to clarify it.**

Line 161: "France to Germany due to different systems for odorized gas (Entsog, 2022c, b)." Presumably, 'odorized' refers to safety requirements, e.g. different countries impose different chemical addition requirements that serve to inhibit opposite exchanges?

**Yes, the purpose of gas odorization is to make the gas smell so that one can detect it because, in its natural state, natural gas is odorless. However, the difference among the odorization systems might be possible to resolve in a short or middle term.**

**The related information can be found in reference *ENTSOG Summer Supply Outlook 2022*, which is already in our reference. In this report, the French and German technicians are exploring both the short-term and long-term solutions for reverse flow scenarios mainly for the storage filling. The report also points out that '*the short-term preparedness can allow for a more even distribution of the storage filling levels at the end of injection period, so that most Central Eastern Europe countries can show the same level of preparedness for the next winter*'. Hence, the reversion of the pipeline flow could be a short-term technician's issue, however, the pipeline capacities could not be solved without massive infrastructures.**

**(For the following comment) This report is the basis for our simulation network and redirection simulation. In the redistribution simulations, we allowed the bidirectional flow for all the pipeline assuming the maximal capacities of remains unchanged. Therefore, we use bi-directional edges for the network (even if the current capacity is 0). For example, the capacity from Germany to France is 614 GWh/d reported by ENTSO-G, however, the capacity from France to Germany is 0 because of the gas odorization. In the gas redirection simulation, the capacities for the two edges, France -> Germany, and Germany -> France are both 614 GWh/d.**

Lines 162, 163: Authors raise interesting potentially-troubling factor: flow irreversibility. Readers needs better understanding of magnitude and frequency of this barrier? Occurs at every 'upstream' (back toward Russia) node? Only in flows across specific edges? Close look at Figure S2 clearly implies some 'loops': how do or would mono-directional flows work in those cases? Perhaps a serious barrier? Topic gets worthy mention later (e.g. line 270 in Challenges and Uncertainties section (good) but again sans quantification?

**We adjusted the edge capacities for different simulations, and as mentioned above, those edges that are unavailable in the current net could be created with short-term or long-term adjustments for the odorization systems.**

**We performed two simulations, the major simulation is used to evaluate the daily Russian gas share in pipeline, consumption, and storage for each country based on the ENTSO-G pipeline data, which does not require the capacity as there is no flow data from France to Germany in the ENTSO-G data.**

**We also simulated the "redistribution" based on the daily surplus/deficit evaluated from our proposed solutions. In this simulation, the pipeline maximal capacity will be used to calculate the "extra" daily transmission capacities that are available. And here the surplus would not be able to be fully transmitted to other countries if there was no sufficient capacity. Based on this simulation, we can easily find out the bottleneck that limits the redistribution.**

**The whole part will be modified as:**

**_'The gas transmission by the current ENTSOG gas pipeline network can be mono-directional between some EU countries, which will result in "bottlenecks" for the gas surplus redistributions (Entsog, 2022b). For example, there is a large transmission capacity (614 GWh/day) from Germany to France, however, with zero capacity from France to Germany due to different systems for gas odorization (Entsog, 2022c, b). We simulated the gas redistribution for both the current network and the network that allows bi-directional flow (as shown in Table S1). The bi-directional network was also evaluated as the gas companies have been working on short-term and long-term solutions for reversing the gas flows although there still remain technical uncertainties (Entsog, 2022c).'_**

Line 183, 184: "assumes a daily balance of the pipeline network, which might over simplify gas balancing processes," not clear how this possible oversimplification would impact overall calculations? Could authors provide an estimate? Would these errors prove systematic (e.g this work always lower or higher than validation products) or would they prove random? Valid point but not clear about impact?

**The assumption that the gas from different sources is well-mixed in a transmission point could be close to the real operational situations. However, in our simulation, we simplify the network. We aggregated all the pipelines between countries as one edge, and points inside the country together as one node. This simplification will result in the well-mixed assumption that might not be achievable, i.e., "over simplified". However, the uncertainty of this simplification would be difficult to evaluate as it largely depends on the operational system and strategy (country-based), the daily imports, transmissions, storage, and consumption conditions (varied daily).**

Line 184, item 3): awkward, not clear what the authors intend?

**The detailed sectoral consumption data for natural gas are not provided by ENTOS-G. Thus we used the consumption variation pattern and energy balance from other data sources to disaggregate the sectoral consumption.**

**The text will be modified as:**

*'our estimation of sectoral consumption might not be able to reproduce unusual daily consumption variations as our values were estimated based on daily temporal total consumption variation patterns from ENTSO-G and monthly (thus smoothed) sectoral Eurostat energy balance to attribute total consumption to each sector.'*

Lines 186-188: valid concern, credit the authors for mentioning social geopolitical issues. Again, do authors expect these to prove negligible? Serious? Disqualifying? Pertinent to this topic but hardly unique to this particular approach?

**Those social geopolitical issues are very important and cannot be termed as negligible to achieving the solutions proposed in this research. However, the focus of this paper is not on those issues, we want to first analyze the *potential* to fill the Russian gas gap assuming e.g., no geopolitical and market limitations. We are considering writing another study that focuses on discussions for those issues based on the data from this study (ESSD is primarily a dataset descriptor journal).**

**We will add text:**

*'…, although they are important yet not in the scope of this study.'*

Line 194: a bit confusing, I think authors mean 'largest consumers of Russian gas'?

**We will replace "biggest" with "largest".**

Line 201, 202: countries with "large" shares but with small domestic consumption presumably pass through much of their RU gas? E.g. large imports but equally large exports? How does reader check this in EUGasSG? Not clear. One needs both EUGasSG and EUGasRP?

**This assumption is true for 'tube' countries with large imports and exports, but not for all the other countries. For example, Nordic countries have large Russian gas shares but low domestic consumption, and mostly import their (Russian) gas from Germany. Another example is the Baltic countries that do not connect to any EU countries and receive gas directly from Russia.**

**Therefore, country size, position in the network and energy structures are the most important factors. On the other hand, EUGasSC provides the gas source of domestic gas consumption within the country based on our simulation, which already considers the "pass-through" of gas across each 'tube' country.**

Line 205: Necessity to group like countries very clear, but this represents another further source of uncertainty? Not itemized above or below?

**No, this step will not provide any extra uncertainties as all the analyses were done for each country (EUGasRP). This grouping step simply adds the results together in order to simplify the visualizations, we do not want to present too many bars in the figures.**

Line 215 and following: Interesting approach. Readers must assume, or need assurance (?) that Dutch TTF price data are open and easily accessible? These data merit listing in text and figure itemizations of important data sources?

**Yes, the daily TTF price is openly accessible, it can be downloaded here:**

https://www.investing.com/commodities/dutch-ttf-gas-c1-futures-historical-data

**We will add this data source.**

Line 234, Section 4.4: Good discussion, follows directly (and, necessarily) from prior data collection and analysis, but - as projections - here we really need uncertainties? Authors could preface entire section with a disclaimer about, e.g plus/minus 10%, 20%. Something?

**We actually included the uncertainties in this section with a range of uncertainty. We estimate the potential solutions based on (technically) best and worst scenarios that represent the upper or lower bounds of our potential. Those scenarios are discussed in the manuscript or in the supplement.**

Line 253 This reader misses in this section a summary of strengths, uncertainties, and validations of EUGasSG data? Promote your work? What new have you discovered and provided, with what strengths and weaknesses. Otherwise, this section moves directly to gap analysis without assuring readers that we start from a good fresh skillful basis?

**The uncertainties and validations have been presented in the previous sections. We do not think it is necessary to repeat them here. EUGasSC provides the source of daily sectoral gas consumption for the EU countries. EUGasPR provides potential daily solutions for filling the Russian gas gap. Based on the two new datasets we assess the potentials for filling the Russian gas gap with three different solutions (one increasing the supply, one substituting the power supply to use less gas, one decreasing the demand). These potentials do not account for social, economic and other geopolitical limitations.  This is our major new contribution to bring publicly available scientific data analysis to the Russian gas climate-energy-political nexus.**

**We will add one summary sentence here:**

*'The two datasets for the first-time document the spatial-temporal-sectoral gas supply sources and potential solutions (at the time of the paper publication) from both the demand and supply side that can alleviate the Russian gas shortage in the EU countries, with a relatively high temporal resolution. However, ...'*

Line 254: Fair enough, but applies not only to international LNG sources?

**We estimate the possible boosting of international LNG imports, and argued that the remaining gap would be able to be filled fill by extra LNG imports. We agree with the reviewer that there are many other possible barriers. Yet, we emphasize that the international LNG/gas market would be very important for filling the Russian gas gap.**

**This sentence will be replaced with**

***'…, our estimates do not contend with social, economic, and political barriers, from the international gas/LNG market and other international cooperation.'***

Line 255: "country-dependent" yes, but earlier authors informed readers of necessity of combing groups of countries based on population size and proximity to Russia. Do those earlier groupings no longer apply here? Or, should authors add something about modifying country-by-country dependencies?

**Yes and no. The two groups are created for different purposes and focus. They are related but not the same. For the first group, we combined the countries by the Russian gas share and total gas consumption, which aims to present the supply-consumption patterns. And for the second group (here), we aim to discriminate the difficulties for countries to fill the Russian gas gap. In the second group, we included more factors, i.e., the potential solution related factors, heating reduction, and energy structure.**

Line 260: If (admittedly) we rarely achieve "perfect cooperation", and if even best cooperation might prove vulnerable to distinct within-country events (e.g. strikes), don't these real-world issues introduce a further degree of uncertainty? At this summary of skillful analysis, authors owe readers a word of advice on how much to trust this work? Please do not let readers make their own assumptions?

**As we have reposed above, those factors are very important but not within the scope of this paper in ESSDD, mainly a dataset description and analysis paper. And we will write another paper that focuses on discussions for those issues based on this paper. We already have conclusion sentences (line 290-293) that point out the gap can be filled on paper, however, the uncertainties from social geopolitical issues are really not within the scope.**

EEGasSG easy to download, open and use but many data show 15 or more significant figures? Not believable, particularly given flow and sector uncertainties. This represents a float vs int problem? Unfortunately, authors reduce their credibility by expressing their data to obviously false precisions?

**This is an important comment. The dataset takes the direct outputs from our simulation model, which resulted in more significant figures not necessarily. The significant figures should be kept the same as ENTSO-G pipeline data (KW h with no decimal places) although it would not change the analysis results in our paper as we aggregated the values to TW h.**

**We will update our dataset to make sure there are no precision issues with the significant figures.**

References: Confusing appearance? Some names in all-caps (e.g. 370, 372), others standard? Not clear the alphabetical organization in too many cases, e.g. Line 343 why Nuclear … here, Line 382 why 'EDF …' here, etc.

**We will double-check the reference format probably with the help of ESSD proofreaders, and make sure there will be no typos in references.**

Line numbers absent in Supplement so hard to comment. But, under Supplement 'Power sector' section, reader finds (two lines below): analyze the diurnal capacities (75% and 95% as moderate and severe cases) for those alternative electricity sources. Properly, 'diurnal' refers to daily as 'nocturnal' refers to nightly; for a full 24-hour period the authors should specify 'diel'? Not a common usage but more correct than current?

**We will add line numbers to the supplement.**

**We will replace "diurnal" with "diel".**

---

## Author Comment (AC2)

Dear reviewer,

Thank you very much for your affirmation of our work as well as your valuable comments.

Here are our responses and potential modifications based on your comments. The text in bold and italics is our response.

**General comment of appreciation**

This paper is not only documenting timely information but entails also important messages for policymakers to understand different options to address the current energy crisis with options at a given cost, also in terms of GHGs. This paper is recommended to receive extra advertisement of ESSD for two reasons. Firstly, it illustrates the importance of data and observation-based evidence for informed decision making. Substitutes for Russian gas supply are needed now and this paper comes with very timely advice. Secondly, science and data sharing can help to keep an open dialogue between many countries in troubled times. It is remarkable how much regional data has been taken up in this exercise.

***Thanks again for your affirmation of our work. With your valuable suggestions, we can deliver a better dataset and information on the EU gas supply-consumption and potential solutions for the current gas crisis related to the Russian supply.***

**Comment on the data used for section 2.3.3:**

It is difficult to assess if the data and information collection is complete for Europe, given the very different approaches in the different EU27 MS. Import from North Africa might benefit extra attention. As an example, ENI Italy was already end April seeking to untap North-Africa's potential for gas supply, getting deals with Algeria and Egypt.

Ref: https://www.al-monitor.com/originals/2022/04/italys-eni-tap-north-african-gas-europe

***This is an important suggestion, we will add one overview figure that summarizes the potential increases of international gas supply to EU to supplementary information or manuscript. And, indeed, the North African countries play important roles (potentially filling about 6~15% of the Russian gas gap) to alleviate the Russian gas deficit by increasing their exports to the EU. Egypt was not fully considered in our current analysis, and we will add Egypt data and discussions to our revised version based on the reference.***

**Comment on the substitution discussed in section 2.3.2:**

Why is biogas production within EU27 not taken up? Even if the source is small, a little upscaling is still possible and all small bits help.

Ref: https://www.euronews.com/green/2022/03/30/biogas-made-from-farm-waste-could-replace-russian-fossil-fuels-in-germany#:~:text=Biogas%20made%20from%20farm%20waste%20could%20replace%20R

ussian%20fossil%20fuels%20in%20Germany,-
Aerial%20view%20of&text=As%20Germany%20attempts%20to%20reduce,as%20an%20al
ternative%20energy%20source.

*We do evaluate the potential capacity from power generated from the biomass sector. But for more detailed sector categories, such as biogas, although they might be alternatives for natural gas, we did not find a solid dataset (also see our response below for the EuroStat) or approach to evaluate their capacity for now. Those might be a topic for our future works. We will add some discussions about those promising alternatives to our paper.*

**Comment on Supplementary Information – gas network simulation**

The total amount of the gas pipelines from North Africa seems reduced. This might be acceptable, if the total volumetric flow rate is fully taken up. In fact, there are three pipelines available from North Africa to Italy: the Transmediterranian (via Tunesia – El Haouaria), the Greenstream (Libya – Mellitah) and the Galsi (Algeria – Koudiet Draoucha) and there are two pipelines available from Algeria to Spain: the Medgaz (Algeria – Beni Saf) and Maghreb-Europe gas (via Morocco – Gibraltar)

Ref: doi:10.1007/s10479-020-03659-9

*We manually collected all the importing points from the ENTSOG map. The following is a table that compares the ENTSOG points from North Africa we used for collecting gas transmission data with the pipelines you suggested. The data for Koudiet Draoucha is currently unavailable from ENTSOG, which might cause the underestimation. Please check more information here:*
*https://transparency.entsog.eu/#/map*

|  | ENTSOG Point Name | Point EicCode |
|---|---|---|
| Tunesia – El Haouaria | MAZARA DEL VALLO | 21Z000000000080V |
| Libya – Mellitah | GELA | 21Z000000000149L |
| Algeria – Beni Saf | ALMERÍA | 21Z0000000002131 |
| Morocco – Gibraltar | TARIFA | 21Z000000000045X |
| Algeria – Koudiet Draoucha | KOUDIET EDDRAOUCH | Planned |

**Comment on Supplementary Information – sectoral splitting validation**

It would benefit the transparency, if EuroSTAT sectors 4 "Industry – E and NE" and sector 5 "sum of the other four sectors" could be described shortly. It would be interesting to understand if "fertiliser production" is part of sector 4 as Industry NE or of sector 5.

*Those terms are directly from EuroStat dataset, we will add explanations for them. In brief, "FC" refers to the final consumption, "Industry – E and NE" refers to the energy and non-energy (used as raw material) consumption in the industrial sector. And the other sectors mainly include energy and non-energy consumption in transport, agriculture, and fisheries. It would be interesting to add discussion on biogas,*

*unfortunately, we did not find any sector directly related to biogas from the gas energy balance dataset (EuroStat, nrg_cb_gas).*

**Minor editorial comments:**

l.2 + l.36: please replace "dramatically" by "drastically"

*Thanks. We will replace "dramatically" with "drastically".*

l.35: More countries saw a sudden decrease and this sentence might be completed with a date and a reference.

*This is background information for 2020 (before the war) which has been noted at the beginning of the sentence.*

l.25 + l.62: when mentioning USA, Australia and Norway, adding North Africa could be appropriate

*We will add more discussion about North Africa as discussed in the previous response.*

l.179: please specify the reason why gas consumption for Latvia and Estonia are underestimated.

*This refers to lines 168-171, the data comparisons among our dataset, Eurostat, and Bp report. For the country-based comparisons (Fig 3e), we find a large underestimate for Latvia-Estonia (-153%) probably due to the incomplete consumption data in ENTSOG for the two countries.*

l.246: adding North Africa could be appropriate.

*We will add North Africa as discussed in the previous response.*

l.264: first verb "could" seems too much for good reading

*We will delete the first "could".*

fig.1: the "legend" figures for input dataset, output dataset and model estimation is not needed and could be deleted, if the first green box reads "Open input dataset"

*Good suggestions. We will edit the title for each box and delete the legend to simplify the figure, i.e., "Open Dataset" to "Open Input Dataset", "Gas Supply Source" to "Gas Supply Source (EUGasSC dataset)", "Potential Solutions" to "Potential Solutions (EUGasRP dataset)".*

fig.2: a) I would prefer to read "annual consumption" instead of "total consumption"

*We did not use "total consumption" for fig.2, are you referring the fig.3a?*

*If so, we will change the "total consumption" to "annual consumption" in fig.3a.*

Fig.5: please write TTF in full

*We will add the full name for TTF, which is Title Transfer Facility.*

Fig.6: this is the difference of two shares, but for which the total maximum is different because of different years. I would prefer to see the difference between 2021 and 2020 (in absolute terms of Russian gas supplied) divided by the total in 2021 (of gas supplied). This would present the change in Russian supply share for 2021 compared to a previous year in a cleaner way. And since 2020 is a special COVID year, a fourth row would be welcome, where also the difference between 2021 and 2019(in absolute) divided by the total in 2021 is also presented.

*Yes, it would be interesting to separate the comparisons for the COVID period. We will replace the third row with two new rows that compare the difference between 2021 and 2020, 2021 and 2019, respectively.*

Fig.7: excellent graph!

*Thanks!*